# STT: Towards Training-Free Zero-Shot Composed Image Retrieval via Semantic Transition and Transportations

## Abstract

Training-free zero-shot composed image retrieval (ZS-CIR) models are recently gaining increasing research interest due to their generalizability and flexibility in unseen multimodal retrieval. Recent LLM-based advances focus on generating the expected target caption by exploring the compositional ability behind the LLMs. Although efficient, we find that 1) the generated captions tend to introduce unexpected features from the reference image due to the semantic gap between the input image and text modification, where the image contains much more details than the text; 2) the point-to-point alignment during the retrieval stage fails to capture diverse compositions. To address these challenges, this paper introduces a novel **S**emantic **T**ransition and **T**ransportation (STT) framework for training-free ZS-CIR tasks. Specifically, given the composed caption inferred by an LLM, we aim to refine it through a transition vector in the embedding space and make it closer to the target image. Combining LLMs with user instruction, the refined caption concentrates more on the core modification intent and thus filters out unnecessary noise. Moreover, to explore diverse alignment during the retrieval stage, we model the caption and image as discrete distributions and reformulate the retrieval task as a set-to-set alignment task. Finally, a bidirectional transportation distance is developed to consider fine-grained alignments across modalities and calculate the retrieval score. Extensive experiments and ablations demonstrate that our method can be general, effective, and beneficial for many CIR tasks.

## 1 Introduction

Composed Image Retrieval (CIR) aims to search for a target image using a compositional query of a reference image and text modification Vo et al. (2019b); Lee et al. (2021); Hosseinzadeh & Wang (2020); Chen et al. (2020b); Baldrati et al. (2022). One of the key challenges is to model the multimodal relationship of the triplet: *<reference image, text modification, target image>*. Previous studies have focused on fusing the input image and modifications within a shared embedding space in a supervised manner Vo et al. (2019a); Delmas et al. (2022); Anwaar et al. (2021a). Generally, these models typically rely on expensive manually-annotated triplets and often exhibit suboptimal performance in unseen scenarios Baldrati et al. (2023); Karthik et al. (2023). Motivated by the success of textual inversion in image generation Gal et al.; Mokady et al. (2023), recent studies have proposed Zero-Shot Composed Image Retrieval (ZS-CIR) Saito et al. (2023); Zeng et al. (2023); Jiang et al. (2024). These models focus on training a mapping network to convert the reference image into continuous textual embeddings and then merge them with text modifications using static templates for target captions, enabling CIR without explicit supervision. Unfortunately, these models also need image-caption pairs to learn the mapping parameters, and the mismatch between textual inversion and CIR may hamper their ability to accurately infer the implicit user intent conveyed in the text modification.

Alternatively, training-free approaches paired with foundation models can achieve effective CIR without additional training and offer improved reasoning capabilities Karthik et al. (2023); Tang et al. (2024a). There are mainly two directions: two-stage methods typically require an image captioner and an LLM to first generate detailed captions of the reference image and then fuse them with text modification via an LLM to produce the target descriptions; one-stage methods unify

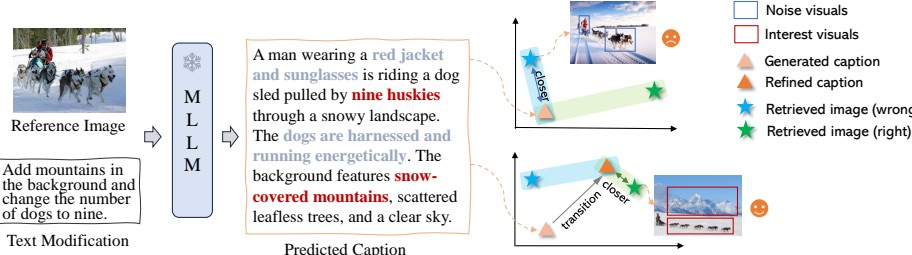

Figure 1: Motivation of our proposed model. Predicted captions from MLLMs typically consist of expected ground-truth sentences (red words) and unexpected visual details (gray words).

this process by employing an MLLM to directly output the target captions given the multimodal queries. Despite considerable progress, several challenges remain. First, the above generation-then-retrieval pipeline is prone to *Extraneous Cognitive Load* Sweller (1988). Specifically, the reference image may trigger information leakage, which in turn leads to overemphasis on irrelevant details, affecting the retrieval performance. As shown in Fig.1, the target caption generated by MLLMs includes extraneous elements such as "red jacket and sunglasses", which are unrelated to the textual modification. Therefore, distracting from the core intent, it diminishes the ability to identify key information, such as "snow-covered mountains". Second, most existing models generate either a single description or simply average multiple descriptions to obtain the final representation Tang et al. (2024a); Yang et al.. However, as one image is worth a thousand words, such point-to-point alignment focuses on partial features and fails to capture complex relations. This inherent heterogeneity between visual and textual representations inevitably leads to semantic imbalance across modalities, leading to suboptimal retrieval prediction Zhu et al. (2024); Chen et al. (2023); Wang et al. (2023).

To address the above issues, this work proposes STT, a novel one-stage, training-free ZS-CIR framework that improves the existing generation-then-retrieval pipeline by introducing **S**emantic **T**ransition and **T**ransportation. Like previous works Tang et al. (2024a), we explore the in-context learning of MLLMs and directly query an MLLM to generate the target caption given the reference image and text modification. Importantly, to address the above asymmetry issues, STT utilizes the uncertainty ability of the language decoder and views the description as a discrete distribution by generating multiple candidates. Each candidate in the distribution focuses on a specific composition pattern, and they together provide a comprehensive understanding of the given query input.

Since reference images may inevitably introduce irrelevant information into captions generated by MLLMs, we propose guiding the textual caption toward the target image via a transition vector in the embedding space (as seen in Fig. 1). Intuitively, an ideal transition vector should bridge the semantic gap between the generated caption and the target image. Here, we aim to solve it in a training-free manner and estimate the transition vector by feeding the text modification into the CLIP text encoder. For one thing, since both the text modification and the generated caption share the same modality, the former can seamlessly refine the latter without introducing a modality gap or requiring extra parameters. For another, the text modification encapsulates the incremental, high-quality, and dense information that shifts from the reference image to the target image, guiding the model to refocus on core semantic information. This transition operates directly in the embedding space—simple yet efficient—and ensures that the final target caption retains diversity while reducing distortion, all of which is highly relevant and beneficial to the retrieval process.

After obtaining high-quality and diverse features of the target captions, it is crucial to align them more effectively with the target images in the embedding space. Similar to the textual domain, STT also models the target image as a discrete distribution by augmenting the image multiple times, where each augmentation captures local visuals of the target image, and they together provide a rich visual representation from the visual domain, facilitating fine-grained alignments in the retrieval process. Finally, a novel bidirectional transport distance is further developed to calculate the similarity of two discrete distributions across the vision-language modalities. Specifically, given the cost matrix that measures the transport cost between the captions and image augmentation, STT designs both a forward path and a backward path to calculate the transport distance from the caption set to the target image set and that from the target image set back to the caption set, respectively. This formulation effectively uncovers fine-grained cross-modal correlations by minimizing the bidirectional transport cost between modalities.

In summary, the proposed STT shares the following contributions:

- We propose a novel one-stage, training-free framework that considers both semantic transition and transportation for ZS-CIR. By introducing the modification-driven transition to the generated caption, the feature drift caused by additional irrelevant noise is compensated in the embedding space.

- We elegantly transform the retrieval procedure into a bidirectional transport problem. Explicitly explore fine-grained alignments between diverse refined textual captions and enhanced target images.

- Extensive comparisons and ablations on four benchmarks demonstrate the effectiveness of the proposed STT with competitive performance in all settings.

## 2 RELATED WORK

### 2.1 COMPOSED IMAGE RETRIEVAL

Composed Image Retrieval (CIR) has inspired various architectural innovations Vo et al. (2019b); Chen et al. (2020b); Lee et al. (2021). Early methods adopt a fusion paradigm to learn joint embeddings of reference image and modification features via contrastive or attention-based objectives Chen & Bazzani (2020); Anwaar et al. (2021b). Some recent works train large-scale retrieval models on millions of web-mined tripletsZhang et al. (2024), but reliance on supervision limits scalability. To address this, Zero-Shot CIR (ZS-CIR) enables retrieval without labeled data. For example, Pic2Word Anwaar et al. (2021b) projects image features into a token embedding space, while SEARLE Baldrati et al. (2023) improves alignment via a text inversion network. Another line of work generates synthetic triplets from image-caption pairs using generative models Gu et al. (2023).

With the advent of foundation models, recent studies have tackled CIR in a training-free manner, leveraging their strong contextual understanding. Two main paradigms have emerged: two-stage methods performs reference image captioning and text manipulation separately, while one-stage methods generate target captions directly from multimodal inputs. For example, CIReVLKarthik et al. (2023) initially employs pre-trained captioning models Li et al. (2023a) to generate caption for a given image. Subsequently, it queries an LLM to refine and recompose the caption based on text modifications for text-to-image retrieval. LDREYang et al. (2024b) considers diverse semantics of the CIR and generates diverse captions at the first stage, and then adopts an ensemble strategy to get the final single feature for the multiple captions. OSrCIRTang et al. (2024a) uses MLLMs to infer user intent by directly processing a query pair, guided by a reflective chain-of-thought prompt. However, both are overwhelmed by the rich semantic information from the reference image, which may overshadow the key modifications. Different from existing LLM-based methods, our STT aims to refine the captions after the generation and improve semantic alignments by formulating the retrieval task as the bidirectional transportation problem.

### 2.2 ALIGNMENT VIA TRANSPORT DISTANCE

Recently, Optimal Transport (OT) Villani (2009) has been widely used for aligning distributions in various domains Redko et al. (2019); Zhao & Zhou (2018;?); Lee et al. (2019); Chen et al. (2020a). Unlike traditional distance metrics like Euclidean distance, OT provides a more geometrically nuanced measure that captures structural similarities between distributions. However, it typically requires iterative optimization via the Sinkhorn algorithm Cuturi (2013), which can be time-consuming. To this end, Conditional Transport (CT) considers the transport plan based on the semantic similarity between samples from two distributions bidirectionally Zheng & Zhou (2021). Its flexibility allows seamless integration with deep learning frameworks, offering lower computational complexity and better scalability, resulting in superior performance in recent alignment tasks Liu et al.. For instance, Tian et al. (2023) exploits transferable statistics with CT to refine biased prototypes to capture unbiased statistics within imbalanced query samples. Li et al. (2023b) design a sparse and layer-wise CT framework to enhance interactions between visual patches and textual labels, ensuring higher semantic consistency for multi-label classification. Notably, CT is inherently well-suited for a key step in CIR—aligning multiple captions with target images—especially when multimodal representations are involved. Motivated by this potential, we transform the traditional point-to-point similarity measure into a minimization of CT-based distance from a distributional perspective.

# 3 METHOD

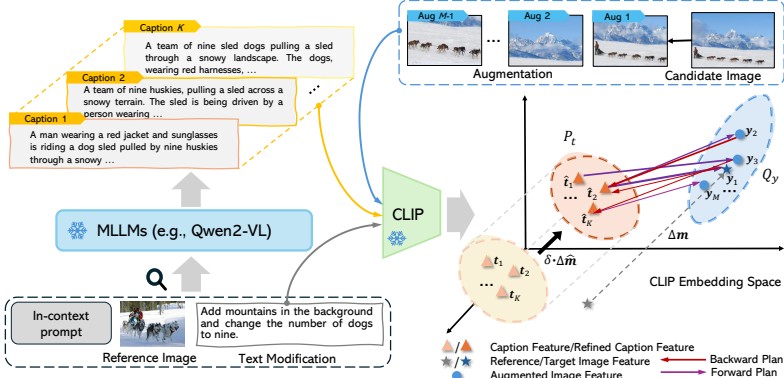

Figure 2: The overall framework of the proposed STT. STT first queries MLLMs to generate multiple captions and then refines them towards the target image via the transition vector. STT models the transferred captions and augmented images as two discrete distributions for fine-grained alignment.

## 3.1 PRELIMINARIES

Let us denote $(x, m, y)$ as the CIR triplet *<reference image, text modification, target image>*, respectively. The first two are multimodal inputs from users describing their retrieval intent. Training-free ZS-CIR aims to search a target image $y$ from an image database $Y = \{y_n\}_{n=1}^{N}$ that satisfies the semantic consistency with both $x$ and $m$, without requiring additional training. Generally, existing approaches follow a generation-then-retrieval pipeline to make the final prediction. They first feed the reference image $x$ and text modification $m$ into a fusing model (such as an MLLM) to obtain the composed description of the target image, denoted as $t = \text{MLLM}(x, m)$. The retrieval score is then calculated by CLIP similarity:

$$p(y = i | x, m) = \frac{\exp(-dis(\boldsymbol{t}, \boldsymbol{y}_i)/\tau)}{\sum_{n=1}^{N} \exp(-dis(\boldsymbol{t}, \boldsymbol{y}_n)/\tau)}, \tag{1}$$

where $\boldsymbol{t} \in R^d$ and $\boldsymbol{y} \in R^d$ are the latent features of $t$ and $y$ in CLIP space, with $d$ denotes the embedding dimension. $dis$ is the distance function and $\tau$ is the temperature parameter. $\boldsymbol{t}$ in Eq. 1 can be viewed as prototypes that capture reasonable visual features of $y$. To find the optimal $\boldsymbol{t}$, recent LLM-based models develop various attempts, including two-stage generation and chain-of-thought reasoning Yang et al. (2024b); Li et al. (2024b); Tang et al. (2024a).

Despite their promising results, We find that 1) the generated description $\boldsymbol{t}$ usually inherits the unnecessary details from the reference image, leading to suboptimal prototype learning; 2) The point estimation of $\boldsymbol{t}$ fails to model complex composed relations, this may limit the uncertainty of $\boldsymbol{t}$ and diminish the generalizability.

## 3.2 SEMANTIC TRANSITION AND TRANSPORTATION

Starting from Eq. 1, we propose a novel training-free ZS-CIR framework to solve the mentioned shortcomings, as illustrated in Fig. 2. Specifically, our proposed model consists of three modules: Querying, Transition, and Alignment. Unlike previous point estimation of $\boldsymbol{t}$, STT views the composed prototype as a discrete distribution, e.g., $P_t$, over the caption space. This allows $P_t$ to focus on various possible target captions, showing higher diversity. To alleviate the issues of unnecessary information pollution, we further transfer the obtained $\boldsymbol{t}$ using the text modification $\boldsymbol{m}$ in the embedding space. $\boldsymbol{m}$ contains high-quality relative information linking the reference image to its target image. Thus, a simple combination strategy is developed to push $\boldsymbol{t}$ to its target image, resulting in more precise $\boldsymbol{t}$. Finally, we also view the target image as a discrete distribution and develop a bidirectional transport distance to align the composed prototypes and target images for fine-grained retrieval.

**Querying.** As discussed above, one of the core challenges is to generate reasonable captions $\boldsymbol{t}$ for image $y$. Inspired by previous works Li et al. (2023a; 2024a), STT aims to solve this with MLLMs

due to their impressive performance in image-text understanding. Concretely, here we explore the in-context learning of MLLM and complete the prompt template: "*<in-context prompt>. <x>. Instruction:<m>. Edited Description:*". Where *<in-context prompt>* helps MLLMs understand the CIR task and output the expected target descriptions. *<x>*, *<m>* are the placeholder of the reference image and text modification.

Intuitively, there are likely several plausible $t$ for each $x$ and $m$ pair, they describe the same target image from different views, To simulate such an ability, we explore the uncertainty generation of MLLMs via sampling from the language model's decoder, to replace the naive greedy decoding used in previous single target description generations. Formally, we combine the top-k and top-p sampling strategy in (Holtzman et al., 2020) and collect $K$ possible target description $t$ with a discrete distribution:

$$P_t = \frac{1}{K} \sum_{k=1}^{K} \delta_{\boldsymbol{t}_k}, \tag{2}$$

where $\delta_{\boldsymbol{t}}$ refers to a point mass located at coordinate $\boldsymbol{t}$, and $\boldsymbol{t}_k$ denotes the text embedding of $k$-th generated description. $P_t$ can be viewed as a semantic set containing $K$ reasonable descriptions, and it thus considers diverse visual features of the target image.

**Transition.** Another challenge comes from the semantic gap between the generated description $t$ and the target image $y$. Generally, an ideal $t$ should highlight the semantic changes while avoiding unnecessary information instructions from the reference image. On one hand, we empirically find that existing LLM-based generators can successfully describe the changed context. On the other hand, they also pay more attention to the reference image due to the limited guidance in text modification $m$. As a result, the output caption usually contains many visual details of the reference image, which act as noise and mislead the retrieval process. To this end, we propose a transition step that pushes the generated captions to the target image at the embedding space. As shown in Fig. 2, let $\Delta\boldsymbol{m}$ denote the difference between $\boldsymbol{x}$ and $\boldsymbol{y}$, it provides incremental semantics from the reference image to its target. Recalling that the text modification $m$ contains high-quality relative information, it is natural to estimate $\Delta\boldsymbol{m}$ using $m$:

$$\Delta\boldsymbol{m} = \boldsymbol{y} - \boldsymbol{x}, \Delta\hat{\boldsymbol{m}} = f(m), \tag{3}$$

where $\boldsymbol{x} \in R^d$ represents the visual embeddings of the reference image $x$ generated by the CLIP image encoder, $f$ is the CLIP text encoder. Once obtaining the relative guidance $\Delta\hat{\boldsymbol{m}}$, $\boldsymbol{t}_k$ can be updated via a simple fusing strategy:

$$\hat{\boldsymbol{t}}_k = (1 - \alpha)\boldsymbol{t}_k + \alpha\Delta\hat{\boldsymbol{m}}, \tag{4}$$

where the first term $\boldsymbol{t}_k$ comes from the MLLMs, and it encodes multimodal knowledge based on the MLLM's understanding of the composed input $(x, m)$. The second term $\Delta\hat{\boldsymbol{m}}$ is derived from the text modification estimation and contains high-quality relative instruction between the reference and target image. The transferred caption $\hat{\boldsymbol{t}}_k$ takes guidance from both directions with a trade-off hyperparameter $\alpha \in [0, 1]$. Now, we can rewrite $P_t$ as: $P_t = \frac{1}{K} \sum_{k=1}^{K} \delta_{\hat{\boldsymbol{t}}_k}$.

**Alignment.** Given the collected discrete distribution $P_t$ in the text domain, we in this section aim to explore the diverse visual features in the image domain with a similar motivation:

$$Q_y = \frac{1}{M} \sum_{m=1}^{M} \delta_{\boldsymbol{y}_m}, \tag{5}$$

where we augment the target image $M - 1$ times and $\{\boldsymbol{y}_m\}_{m=2}^{M}$ are the embeddings of the augmented images. Unlike previous ZS-CIR models that view the target image as a single point, $Q_y$ in Eq. 5 provides us with multiple views of $y$, leading to the following fine-grained retrieval strategy.

Moving beyond the point-to-point similarity measurement in Eq. 1, we here develop a bidirectional distance of two discrete distributions $\mathcal{L}_{bi}(P_t, Q_y)$ under the CT framework. Specifically, $\mathcal{L}_{bi}$ consists of two transport costs: the forward cost that measures the expected transport cost from the reference to the target image and the backward cost that inverses the direction:

$$\begin{aligned} \mathcal{L}_{bi}(P_t, Q_y) &= \mathcal{L}_{P_t \to Q_y}(P_t, Q_y) + \mathcal{L}_{Q_y \to P_t}(P_t, Q_y) \\ &= \sum_{m,k} \pi(\boldsymbol{y}_m|\hat{\boldsymbol{t}}_k)c(\hat{\boldsymbol{t}}_k, \boldsymbol{y}_m) + \pi(\hat{\boldsymbol{t}}_k|\boldsymbol{y}_m)c(\boldsymbol{y}_m, \hat{\boldsymbol{t}}_k), \end{aligned} \tag{6}$$

where the cost function $c(\boldsymbol{y}, \hat{\boldsymbol{t}}) = c(\hat{\boldsymbol{t}}, \boldsymbol{y})$ is specified as the cosine distance to measure the transport cost between points $\hat{\boldsymbol{t}}$ and $\boldsymbol{y}$. $\pi(\boldsymbol{y}|\hat{\boldsymbol{t}})$ denotes the transport plan in the forward path, and it measures how likely $\hat{\boldsymbol{t}}$ will be transported to $\boldsymbol{y}$:

$$\pi(\boldsymbol{y}_m|\hat{\boldsymbol{t}}_k) = \frac{\exp(\hat{\boldsymbol{t}}_k^T \boldsymbol{y}_m/\tau)}{\sum_{m'=1}^{M} \exp(\hat{\boldsymbol{t}}_k^T \boldsymbol{y}_{m'}/\tau)}. \tag{7}$$

Naturally, the closer $\hat{\boldsymbol{t}}$ and $\boldsymbol{y}$ are in the embedding space, the higher the transport probability from $\hat{\boldsymbol{t}}$ to $\boldsymbol{y}$. $\pi(\hat{\boldsymbol{t}}|\boldsymbol{y})$ is defined in a similar way:

$$\pi(\hat{\boldsymbol{t}}_k|\boldsymbol{y}_m) = \frac{\exp(\boldsymbol{y}_m^T \hat{\boldsymbol{t}}_k/\tau)}{\sum_{k'=1}^{K} \exp(\boldsymbol{y}_m^T \hat{\boldsymbol{t}}_{k'}/\tau)}. \tag{8}$$

Mathematically, $\mathcal{L}_{bi}$ in Eq. 6 calculates the distance between two discrete distributions in both forward and backward directions. This benefits the alignment across the vision and language domains, showing better multimodal retrieval ability. Moreover, $\mathcal{L}_{bi}$ views the generated caption and target image as two discrete distributions, which show great potential in modeling diverse semantics.

Once obtain the bidirectional distance between the generated descriptions and the target image, we can rewrite Eq. 1 with $\mathcal{L}_{bi}$, resulting in a more general and fine-grained prediction score:

$$p(y = i|x, m) = \frac{\exp(-\mathcal{L}_{bi}(P_t, Q_{y_i}))}{\sum_{n=1}^{N} \exp(-\mathcal{L}_{bi}(P_t, Q_{y_n}))}. \tag{9}$$

We summarize the whole inference algorithm of STT in the Appendix. 1.

## 4 EXPERIMENTS

### 4.1 EXPERIMENTAL SETUP

**Datasets.** Following previous works Tang et al. (2024a), we evaluate our proposed model on four commonly used CIR datasets , which vary in CIR tasks, image domains and dataset sizes, including CIRR Liu et al. (2021), CIRCO Baldrati et al. (2023), FashionIQ Wu et al. (2021), and GeneCIS Vaze et al. (2023). CIRR is the first natural image dataset for CIR. CIRCO comes from COCO2017 Lin et al. (2014) and has multiple ground truths for each query. FashionIQ focuses on fashion-related retrieval and consists of three subsets: shirt, dress, and toptee. GeneCIS contains images from MS-COCO Lin et al. (2014) and Visual Attributes in the Wild Pham et al. (2021), offering four task variations around objects and attributes. We report the original benchmark metrics for each dataset: *e.g.*, Recall@k(R@k) for CIRR, GeneCIS, and FashionIQ, and mean average precision (mAP@k) for CIRCO due to its multiple labels.

**Baselines.** We compare our STT with recent advances, grouped as training-dependent and training-free models. The former often optimize a mapping network to project the reference image into text tokens, including 1) **Pic2Word** Saito et al. (2023), 2) **SEARLE** Baldrati et al. (2023), 3) **Context-I2W** Tang et al. (2024b), and 4) **LinCIR** Gu et al.. Training-free methods focus more on improving ZS-CIR with large language models, including 5) **CIReVL** Karthik et al. (2023), 6) **LDRE** Yang et al. (2024b), 7) **OSrCIR** Tang et al. (2024a) and 8) **SEIZE** Yang et al. (2024a). Unlike previous training-free methods, the proposed STT aims to refine the composed caption in the embedding space and explore fine-grained alignment across vision-language domains. For all training-based baselines, we directly report the results according to their official papers. For training-free models, to make a fair comparison, we reproduce their results on the FashionIQ dataset according to their released codes and report results on other datasets according to the original papers.

**Implementation Details.** We employ the open-source Qwen2-VL-7B as our MLLM by default, while we also report the results on various MLLMs in Appendix Sec. A.3. To generate diverse descriptions, we follow the similar setting to previous works and apply $\tau = 0.7$, top-k(k=50), top-p(p=0.8) at the querying stage. In terms of the image augmentation, we used only random resized crop and random horizontal flip for each image. We set the number of captions as $K = 5$, the number of augmentations as $M = 10$, and employ a default value of $\alpha = 0.45$. The default retrieval model is CLIP-L/14 from the official OpenAI implementation Radford et al. (2021). We also report the retrieval results of LLM-based models (CIReVL, OSrCIR, and our STT) on CLIP-bigG-14 from the OpenCLIP implementation Cherti et al. (2023) for the analysis of scaling laws. All experiments are conducted on a single NVIDIA A6000 GPU.

Table 1: Performance comparison on CIRCO and CIRR datasets. The top two results are highlighted in **bold** and underline, respectively. More comparisons are reported in the Appendix Tab. 6.

| CIRCO + CIRR → | | | CIRCO | | | | CIRR | | | | | |
|---|---|---|---|---|---|---|---|---|---|---|---|---|
| Metric | | | mAP@k | | | | Recall@k | | | Recall$_{Subset}$@k | | |
| Arch | Method | Train | k=5 | k=10 | k=25 | k=50 | k=1 | k=5 | k=10 | k=1 | k=2 | k=3 |
| ViT-L/14 | Pic2Word | ✓ | 8.72 | 9.51 | 10.64 | 11.29 | 23.90 | 51.70 | 65.30 | 53.76 | 74.46 | 87.08 |
| | SEARLE | ✓ | 11.68 | 12.73 | 14.33 | 15.12 | 24.24 | 52.48 | 66.29 | 53.76 | 75.01 | 88.19 |
| | LinCIR | ✓ | 12.59 | 13.58 | 15.00 | 15.85 | 25.04 | 53.25 | 66.68 | 57.11 | 77.37 | 88.89 |
| | Context-I2W | ✓ | 13.04 | 14.62 | 16.14 | 17.16 | 25.60 | 55.10 | 68.50 | - | - | - |
| | CIReVL | ✗ | 18.57 | 19.01 | 20.89 | 21.80 | 24.55 | 52.31 | 64.92 | 59.54 | 79.88 | 89.69 |
| | LDRE | ✗ | 23.35 | 24.03 | 26.44 | 27.50 | 26.53 | 55.57 | 67.54 | 60.43 | 80.31 | 89.90 |
| | OSrCIR | ✗ | 23.87 | 25.33 | 27.84 | 28.97 | **29.45** | 57.68 | 69.86 | 62.12 | 81.92 | 91.10 |
| | SEIZE | ✗ | 24.98 | 25.82 | 28.24 | 28.35 | 28.65 | 57.16 | 69.23 | 62.22 | 84.05 | 92.34 |
| | **STT(Ours)** | ✗ | **25.55** | **26.27** | **28.81** | **29.99** | 28.87 | **57.97** | **69.90** | **65.22** | **84.10** | **92.37** |
| ViT-G/14 | CIReVL | ✗ | 26.77 | 27.59 | 29.96 | 31.03 | 34.65 | 64.29 | 75.06 | 67.95 | 84.87 | 93.21 |
| | LDRE | ✗ | 31.12 | 32.24 | 34.95 | 36.03 | 36.15 | 66.39 | 77.25 | 68.82 | 85.66 | 93.76 |
| | OSrCIR | ✗ | 30.47 | 31.14 | 35.03 | 36.59 | 37.26 | 67.25 | 77.33 | 69.22 | 85.28 | 93.55 |
| | SEIZE | ✗ | 32.46 | 33.77 | 36.46 | 37.55 | 38.87 | 69.42 | 79.42 | **74.15** | 89.23 | 95.71 |
| | **STT (Ours)** | ✗ | **34.40** | **35.56** | **38.07** | **40.02** | **39.23** | **69.95** | **79.56** | 73.56 | **89.50** | **95.86** |

Table 2: Performance comparison on Fashion-IQ datasets. The top two results are highlighted in **bolded** and underlined. More results with different backbones are reported in Tab. 7.

| Fashion-IQ → | | | Shirt | | Dress | | Toptee | | Average | |
|---|---|---|---|---|---|---|---|---|---|---|
| Arch | Method | Train | R@10 | R@50 | R@10 | R@50 | R@10 | R@50 | R@10 | R@50 |
| ViT-G/14 | Pic2Word | ✓ | 33.17 | 50.39 | 25.43 | 47.65 | 35.24 | 57.62 | 31.28 | 51.89 |
| | SEARLE | ✓ | 36.46 | 55.35 | 28.16 | 50.32 | 39.83 | 61.45 | 34.81 | 55.71 |
| | LinCIR | ✓ | **46.76** | 65.11 | 38.08 | 60.88 | **50.48** | 71.09 | **45.11** | 65.69 |
| | CIReVL | ✗ | 29.85 | 51.07 | 27.07 | 49.53 | 35.80 | 56.14 | 32.19 | 52.36 |
| | OSrCIR | ✗ | 38.65 | 54.71 | 33.02 | 54.78 | 41.04 | 61.83 | 37.57 | 57.11 |
| | SEIZE | ✗ | 43.60 | **65.42** | **39.61** | 61.02 | 45.94 | **71.12** | 43.05 | **65.85** |
| | **STT(Ours)** | ✗ | 39.48 | 56.59 | 35.04 | 56.74 | 42.86 | 64.95 | 39.12 | 59.43 |

Table 3: Performance comparison on GeneCIS datasets. The top two results are highlighted in **bolded** and underlined, respectively.

| GeneCIS → | | | Focus Attribute | | | Change Attribute | | | Focus Object | | | Change Object | | | Average |
|---|---|---|---|---|---|---|---|---|---|---|---|---|---|---|---|
| Arch | Method | Train | R@1 | R@2 | R@3 | R@1 | R@2 | R@3 | R@1 | R@2 | R@3 | R@1 | R@2 | R@3 | R@1 |
| ViT-L/14 | SEARLE | ✓ | 17.1 | 29.6 | 40.7 | 16.3 | 25.2 | 34.2 | 12.0 | 22.2 | 30.9 | 12.0 | 24.1 | 33.9 | 14.4 |
| | LinCIR | ✓ | 16.9 | 30.0 | 41.5 | 16.2 | 28.0 | 36.8 | 8.3 | 17.4 | 26.2 | 7.4 | 15.7 | 25.0 | 12.2 |
| | Context-I2W | ✓ | 17.2 | 30.5 | 41.7 | 16.4 | 28.3 | 37.1 | 17.9 | 26.9 | 7.7 | 16.0 | 25.4 | 12.7 | |
| | CIReV | ✗ | 19.5 | 31.8 | 42.0 | 14.4 | 26.0 | 35.2 | 12.3 | 21.8 | 30.5 | 17.2 | 28.9 | 37.6 | 15.9 |
| | OSrCIR | ✗ | **20.9** | 33.1 | 44.5 | 17.2 | 28.5 | 37.9 | 15.0 | 23.6 | 34.2 | 18.4 | 30.6 | 38.3 | 17.9 |
| | SEIZE | ✗ | 20.5 | 33.4 | 45.0 | 17.6 | 28.9 | 38.5 | 15.4 | 25.6 | 36.2 | 18.7 | 30.9 | 39.8 | 18.1 |
| | **STT(Ours)** | ✗ | 20.3 | **34.6** | **46.4** | **18.3** | **29.8** | **41.6** | **16.8** | **28.5** | **38.4** | **18.8** | **31.0** | **40.3** | **18.6** |
| ViT-G/14 | CIReVL | ✗ | 20.9 | 34.4 | 44.9 | 16.5 | 29.0 | 39.8 | 15.1 | 25.6 | 33.4 | 18.5 | 31.6 | 41.4 | 17.8 |
| | OSrCIR | ✗ | 22.7 | **36.4** | 47.0 | 17.9 | 30.8 | 42.0 | 16.9 | 28.4 | 36.7 | **21.0** | **33.4** | **44.2** | 19.6 |
| | SEIZE | ✗ | **22.9** | 36.2 | 47.3 | 18.6 | 31.4 | 42.7 | 18.2 | 28.8 | 37.6 | 19.6 | 33.0 | 43.5 | 19.8 |
| | **STT (Ours)** | ✗ | 21.9 | **36.4** | **47.9** | **19.6** | **31.9** | **42.8** | **20.2** | 30.3 | **39.6** | 19.7 | 33.2 | 43.4 | **20.4** |

## 4.2 RESULTS ANALYSIS

We run all experiments three times with different random seeds and report the mean value to ensure reliability. We report the comparison on the hidden test set of CIRCO and CIRR datasets in Tab. 1 (All results are obtained from the submission server provided in Baldrati et al. (2023) and Liu et al. (2021)). These two CIR datasets focus on foreground and background differentiation and fine-grained image editing. From the results, we find that our proposed STT achieves the best or second-best results in most cases among baselines, including training-free and textual inversion models. STT underperforms baselines on the CIRR dataset in the case of $k = 1$. This may be due to the noisy annotation in CIRR, where the reference image is less related to the target image Baldrati et al. (2023).

Tab. 2 shows the comparison in the validation set of the Fashion-IQ dataset with ViT-G/14, and more comparisons on other backbones can be found in the Appendix Tab. 7. Interestingly, we find that text inversion-based models (e.g., LinCIR) outperform LLM-based models in the fashion image editing task. This may be due to the fact that most images in Fashion-IQ are relatively simple, containing only a pure background. This may limit the ability of MLLMs, and in contrast, training-dependent models

tend to describe the reference image more correctly. Among LLM-based models, SEIZE achieves the best recall. Both SEIZE and STT refine the target captions, where STT employs modification text as transition vectors, while SEIZE utilizes generated reference captions to adjust final predictions. We find that the modification text in Fashion-IQ is overly simplistic (*e.g.*, is solid white and is a lighter color), resulting in suboptimal guidance for STT. SEIZE relies on a pre-trained caption model to generate reference captions, which may be better suited for Fashion-IQ's simplified scenarios. However, as shown in the Tab. 5, this approach increases the computational cost during inference.

Lastly, we further test the object and attribute composition ability of our model on the GeneCIS dataset, with the results listed in Tab. 3. Unlike previous datasets that provide a detailed text modification sentence, GeneCIS uses single-word instruction to express the user's intent, *e.g.*, focusing/changing a specific object or attribute. From the results, we find that our proposed STT achieves the best results in 19/24 cases and outperforms the others at the average score. On the one hand, STT preserves rich multimodal knowledge to interpret implicit inputs while filtering out noise through transition. On the other hand, it views the caption as a discrete distribution, showing great potential in capturing diverse visual semantics, leading to fine-grained retrieval for this complex task. More comparisons are reported in Appendix Tab. 8.

### 4.3 FURTHER ANALYSIS

In addition to the numerical comparisons on four CIR benchmarks. In this section, we provide further analysis of the ablation results and visualizations of the proposed modules.

**Main component analysis.** To evaluate the impacts of each proposed module in STT, we report the ablation results in Tab. 4. We find that 1) both transition and transportation show a positive improvement compared to the base model (first row). This highlights the motivation of our STT: the unnecessary visual details introduced in original captions and the simple point-based alignment; 2) The transition module achieves higher improvements than the transportation module in most cases. This shows the validity of our proposed transition vector in assessing the difference between the reference and target images. It highlights the text modification and offers useful guidance to steer the original caption toward the target image, resulting in more correct alignments.

Table 4: Ablation results on the transition and transportation modules. All results are conducted on CIRCO and CIRR datasets with CLIP-bigG/14.

| CIRCO + CIRR → | | CIRCO | | | | CIRR | | | | | |
|---|---|---|---|---|---|---|---|---|---|---|---|
| Strategy | | mAP@k | | | | Recall@k | | | Recall$_{Subset}$@k | | |
| *Transition* | *Transportation* | k=5 | k=10 | k=25 | k=50 | k=1 | k=5 | k=10 | k=1 | k=2 | k=3 |
| ✗ | ✗ | 31.23 | 32.87 | 36.32 | 38.04 | 37.22 | 67.36 | 77.84 | 69.93 | 86.48 | 94.05 |
| ✓ | ✗ | 31.89 | 34.46 | 37.94 | 39.67 | 38.33 | 68.45 | 78.03 | 72.81 | 88.13 | 94.51 |
| ✗ | ✓ | 32.14 | 34.78 | 37.87 | 39.48 | 38.48 | 68.38 | 78.34 | 72.15 | 88.04 | 94.48 |
| ✓ | ✓ | **34.40** | **35.56** | **38.07** | **40.02** | **39.23** | **69.95** | **79.56** | **73.56** | **89.50** | **95.86** |

**Impacts of caption number and augmentation views.** We report the study of caption number $K$ and augmentation views $M$ in Eq. 5 on CIRCO dataset, with results in Fig. 3 . From these results, we first find that the performance shows a large improvement when $K > 1$. This demonstrates the effectiveness of using diverse captions, especially after semantic transitions, to capture rich semantic information. Moreover, increasing the number of image augmentations $M$ consistently enhances performance, highlighting the value of multi-scale image diversity. We suggest that setting $K = 5$ and $M = 10$ is sufficient to achieve strong performance across most datasets.

**Impacts of the bidirectional distance.** Recalling that we propose to measure the distance between the modified caption $P_t$ and the target image augmentation $Q_y$ using a bidirectional transport distance in Eq. 6. This enables the proposed STT to not only calculate the transport cost from $P_t$ to $Q_y$ but also consider the reverse cost, showing better alignments across the vision-language domains. To identify such improvements, we replace the bidirectional CT distance with the optimal transport (OT) distance and report the comparison on four datasets in Fig. 4. Note that the CT distance consistently beats the OT distance in all cases, showing the effectiveness of our bidirectional alignment.

**Visualization Analysis.** From the above analysis, we find that the transition module plays a key role in improving the CIR performance. It estimates the relative vector as the difference between the reference and target image and provides correct information to refine the generated captions. To make a clearer understanding, we visualize the heat maps of two samples from the GeneCIS dataset on the "Focus Object" task in Fig. 5. Here, we directly calculate the cosine similarity between the

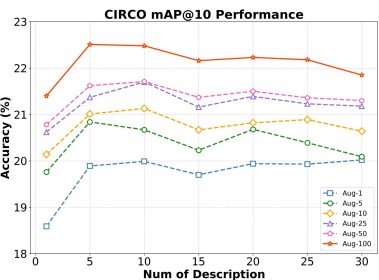

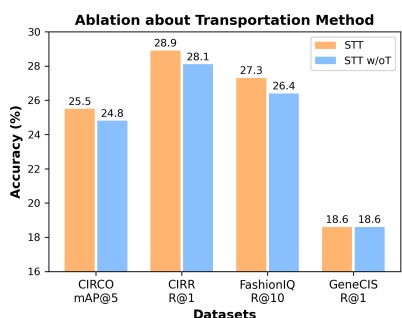

Figure 3: Ablation on the number of descriptions and image augmentations (Aug-$m$:$m$ augmentations per image) on the CIRCO dataset with CLIP-B/32.

Figure 4: Ablation results of different alignment strategies across four datasets with CLIP-L/14.

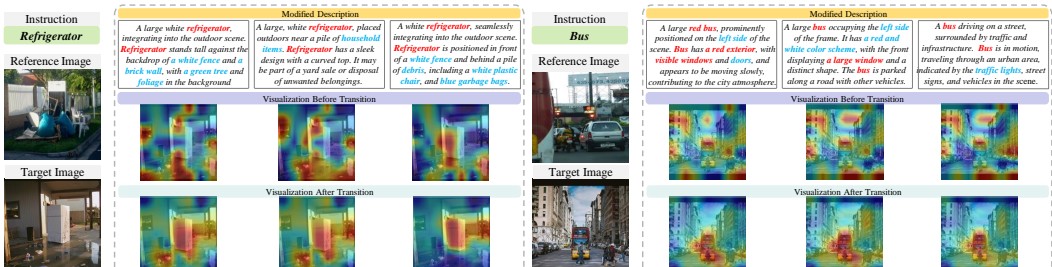

Figure 5: Visualization of the GeneCIS dataset on the 'Focus Object' task. Heatmaps before and after the transition on target image are shown. Captions generated by MLLMs often contain irrelevant visual noise (blue text), while the STT model effectively suppresses such noise and highlights the correct focus object (red text).

Table 5: Efficiency comparison with SOTA methods. Inference times (seconds per query) are reported for each model using a single NVIDIA A6000 GPU.

| Model | Pic2Word | SEARLE | CIReVL | OSrCIR | LDRE | SEIZE | STT |
|---|---|---|---|---|---|---|---|
| **Time (s)** | <0.01 | <0.01 | 2.97+0.85=3.82 | 6.65 | 1.30+2.98=4.28 | 5.8+4.5=10.3 | 3.5 |

visual patch embeddings and the caption embedding as the score of the heat maps. The first row denotes three generated captions and the last two heat maps denote the corresponding visualization results before and after the transition module. We find that the generated captions indeed introduce visual noise for the CIR task. For example, the retrieval attentions are often disturbed by visual noise such as "white fence", "white color" and "left side of the scene". In contrast, the visualizations of STT often focus on the correct object, leading to higher CIR performance.

**Efficiency Analysis.** Table 5 shows the inference times (seconds per query) of various methods. Overall, LLM-based methods generally incur higher test-time latency than traditional mapping models due to the query stage. For instance, CIReVL, LDRE and SEIZE are two-stage approaches that require significant time to generate target descriptions, while OSrCIR involves extensive chain-of-thought reasoning during inference. In contrast, our STT achieves the lowest test time among LLM-based methods, showing the efficiency of our proposed modules.

## 5 CONCLUSION

We propose STT, a novel one-stage, training-free framework for the zero-shot composed image retrieval task. STT improves the quality of the generated caption by MLLMs via a transition vector and views the captions and target image as two discrete distributions in the embedding space. A bidirectional transport distance is developed to measure the similarity across the vision-language domains. Our approach not only achieves strong performance on four CIR benchmarks but also, provides interpretability via the visualization of the transferred caption and the target images. Extensive ablations also demonstrate the effectiveness of the proposed modules.

REPRODUCIBILITY STATEMENT

We provide detailed descriptions of the model, training procedure, and evaluation in the main text. Additional implementation details, hyperparameters, and ablation studies are included in the Appendix.

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

# A APPENDIX

## A.1 ADDITIONAL COMPARATIVE RESULTS.

We in this section included more comprehensive comparisons with more methods across various architectures on all datasets presented in Tab. 6, Tab. 7, and Tab. 8. It should be noted that in Tab. 7, the notation (*) indicates that we reproduced the experiments using the OpenAI weights, and the (†) indicates that we reproduced the experiments using the OpenCLIP weights, respectively. From these comparisons, our approach outperforms all the baselines in most cases, showing the efficiency of STT's three operations.

Table 6: Performance comparison on CIRCO and CIRR datasets. Both ViT-B and ViT-L are loaded from OpenAI official weights, while ViT-G is loaded from OpenCLIP.

| CIRCO + CIRR → | | | CIRCO | | | | CIRR | | | | | |
|---|---|---|---|---|---|---|---|---|---|---|---|---|
| Metric | | | mAP@k | | | | Recall@k | | | Recall$_{Subset}$@k | | |
| Arch | Method | Train | k=5 | k=10 | k=25 | k=50 | k=1 | k=5 | k=10 | k=1 | k=2 | k=3 |
| ViT-B/32 | PALAVRA | ✓ | 4.61 | 5.32 | 6.33 | 6.80 | 16.62 | 43.49 | 58.51 | 41.61 | 65.30 | 80.94 |
| | SEARLE | ✓ | 9.35 | 9.94 | 11.13 | 11.84 | 24.00 | 53.42 | 66.82 | 54.89 | 76.60 | 88.19 |
| | CIReVL | ✗ | 14.94 | 15.42 | 17.00 | 17.82 | 23.94 | 52.51 | 66.00 | 60.17 | 80.05 | 90.19 |
| | LDRE | ✗ | 17.96 | 18.32 | 20.21 | 21.11 | 25.69 | 55.13 | 69.04 | 60.53 | 80.65 | 90.70 |
| | OSrCIR | ✗ | 18.04 | 19.17 | 20.94 | 21.85 | 25.42 | 54.54 | 68.19 | 62.31 | 80.86 | 91.13 |
| | SEIZE | ✗ | 19.04 | 19.64 | 21.55 | 22.49 | **27.47** | **57.42** | 70.17 | 65.59 | **84.48** | 92.77 |
| | **STT(Ours)** | ✗ | **20.26** | **21.01** | **23.01** | **24.04** | 25.83 | 55.25 | **70.20** | **65.64** | 83.60 | **92.80** |
| ViT-L/14 | Pic2Word | ✓ | 8.72 | 9.51 | 10.64 | 11.29 | 23.90 | 51.70 | 65.30 | 53.76 | 74.46 | 87.08 |
| | SEARLE | ✓ | 11.68 | 12.73 | 14.33 | 15.12 | 24.24 | 52.48 | 66.29 | 53.76 | 75.01 | 88.19 |
| | LinCIR | ✓ | 12.59 | 13.58 | 15.00 | 15.85 | 25.04 | 53.25 | 66.68 | 57.11 | 77.37 | 88.89 |
| | Context-I2W | ✓ | 13.04 | 14.62 | 16.14 | 17.16 | 25.60 | 55.10 | 68.50 | - | - | - |
| | CIReVL | ✗ | 18.57 | 19.01 | 20.89 | 21.80 | 24.55 | 52.31 | 64.92 | 59.54 | 79.88 | 89.69 |
| | LDRE | ✗ | 23.35 | 24.03 | 26.44 | 27.50 | 26.53 | 55.57 | 67.54 | 60.43 | 80.31 | 89.90 |
| | OSrCIR | ✗ | 23.87 | 25.33 | 27.84 | 28.97 | **29.45** | 57.68 | 69.86 | 62.12 | 81.92 | 91.10 |
| | SEIZE | ✗ | 24.98 | 25.82 | 28.24 | 28.35 | 28.65 | 57.16 | 69.23 | 62.22 | 84.05 | 92.34 |
| | **STT(Ours)** | ✗ | **25.55** | **26.27** | **28.81** | **29.99** | 28.87 | **57.97** | **69.90** | **65.22** | **84.10** | **92.37** |
| ViT-G/14 | Pic2Word | ✓ | 5.54 | 5.59 | 6.68 | 7.12 | 30.41 | 58.12 | 69.23 | 68.92 | 85.45 | 93.04 |
| | SEARLE | ✓ | 13.20 | 13.85 | 15.32 | 16.04 | 34.80 | 64.07 | 75.11 | 68.72 | 84.70 | 93.23 |
| | LinCIR | ✓ | 19.71 | 21.01 | 23.13 | 24.18 | 35.25 | 64.72 | 76.05 | 63.35 | 82.22 | 91.98 |
| | CIReVL | ✗ | 26.77 | 27.59 | 29.96 | 31.03 | 34.65 | 64.29 | 75.06 | 67.95 | 84.87 | 93.21 |
| | LDRE | ✗ | 31.12 | 32.24 | 34.95 | 36.03 | 36.15 | 66.39 | 77.25 | 68.82 | 85.66 | 93.76 |
| | OSrCIR | ✗ | 30.47 | 31.14 | 35.03 | 36.59 | 37.26 | 67.25 | 77.33 | 69.22 | 85.28 | 93.55 |
| | SEIZE | ✗ | 32.46 | 33.77 | 36.46 | 37.55 | 38.87 | 69.42 | 79.42 | **74.15** | 89.23 | 95.71 |
| | **STT (Ours)** | ✗ | **34.40** | **35.56** | **38.07** | **40.02** | **39.23** | **69.95** | **79.56** | 73.56 | **89.50** | **95.86** |

## A.2 ALGORITHM OF STT PROCESS

We summarize the detailed inference algorithm of STT in Alg. 1.

---

**Algorithm 1:** Inference algorithm of STT.

---

**Input**: reference image $x$, text modification $m$, target image database $Y = \{y_n\}_{n=1}^{N}$, a pre-trained CLIP model, and a pre-trained MLLMs. The number of query times $K$, and the number of image augmentations $M$.

**Output**: The retrieval score $p(y|x, m)$ over all target images. **Querying**: Complete the input prompts with $x$ and $m$, and query MLLM K times to collect the descriptions $P_t$ from Eq. 2.

**Transition**: Calculate $\Delta m$ in Eq. 3 by feeding $m$ into CLIP text encoder, and then obtain the transferred $P_t$ from Eq. 4.

**Alignment**: Collect $Q_y$ in Eq. 5 by augmenting target image $y_n$ for $M-1$ times, and then calculate $\mathcal{L}_{P_t, Q_{y_n}}$ from Eq. 6. image $y_n$ in Y Calculate $\mathcal{L}_{P_t, Q_{y_n}}$ according to **Alignment** step.

**Return** Calculate the retrieval score from Eq. 9.

---

Table 7: Performance comparison on Fashion-IQ datasets. Both ViT-B and ViT-L are loaded from OpenAI official weights, while ViT-G is loaded from OpenCLIP. (*) denotes we rerun the experiments on the OpenAI weights, and (†) denotes we rerun the experiments on the OpenCLIP weights.

| Fashion-IQ → | | | Shirt | | Dress | | Toptee | | Average | |
|---|---|---|---|---|---|---|---|---|---|---|
| Arch | Method | Train | R@10 | R@50 | R@10 | R@50 | R@10 | R@50 | R@10 | R@50 |
| ViT-B/32 | PALAVRA | ✓ | 21.49 | 37.05 | 17.25 | 35.94 | 20.55 | 38.76 | 19.76 | 37.25 |
| | SEARLE | ✓ | 24.44 | 41.61 | 18.54 | 39.51 | 25.70 | 46.46 | 22.89 | 42.53 |
| | CIReVL | ✗ | 28.36 | 47.84 | 25.29 | 46.36 | 31.21 | 53.85 | 28.28 | 49.35 |
| | CIReVL* | ✗ | 22.03 | 37.00 | 13.34 | 30.14 | 18.97 | 38.19 | 18.11 | 35.11 |
| | CIReVL† | ✗ | 27.72 | 46.12 | 22.01 | 41.60 | 30.09 | 52.22 | 26.60 | 46.64 |
| | OSrCIR | ✗ | 31.16 | 51.13 | 29.35 | 50.37 | 36.51 | 58.71 | 32.34 | 53.40 |
| | OSrCIR* | ✗ | 22.77 | 40.87 | 17.01 | 37.04 | 20.75 | 41.00 | 20.18 | 39.6 |
| | OSrCIR† | ✗ | 32.83 | 52.06 | 29.75 | 51.91 | 36.31 | 58.24 | 32.96 | 54.07 |
| | SEIZE | ✗ | 29.38 | 47.97 | 25.37 | 46.84 | 32.07 | 54.78 | 28.94 | 49.35 |
| | **STT(Ours)** | ✗ | 25.22 | 44.16 | 18.59 | 40.16 | 25.97 | 47.61 | 23.26 | 43.98 |
| ViT-L/14 | Pic2Word | ✓ | 26.20 | 43.60 | 20.00 | 40.20 | 27.90 | 47.40 | 24.70 | 43.70 |
| | SEARLE | ✓ | 26.89 | 45.58 | 20.48 | 43.13 | 29.32 | 49.97 | 25.56 | 46.23 |
| | LinCIR | ✓ | 29.10 | 46.81 | 20.92 | 42.44 | 28.81 | 50.18 | 26.28 | 46.49 |
| | Context-I2W | ✓ | 29.70 | 48.60 | 23.10 | 45.30 | 30.60 | 52.90 | 27.80 | 48.90 |
| | CIReVL | ✗ | 26.01 | 44.76 | 24.79 | 44.76 | 31.36 | 53.65 | 27.39 | 47.72 |
| | CIReVL* | ✗ | 24.34 | 40.28 | 14.68 | 32.62 | 23.41 | 41.97 | 20.81 | 38.29 |
| | CIReVL† | ✗ | 28.85 | 45.78 | 22.16 | 41.35 | 30.85 | 51.25 | 27.29 | 46.12 |
| | OSrCIR | ✗ | 33.17 | 52.03 | 29.70 | 51.81 | 36.92 | 59.27 | 33.26 | 54.37 |
| | OSrCIR* | ✗ | 27.58 | 44.31 | 18.69 | 39.02 | 25.80 | 46.00 | 24.02 | 43.11 |
| | OSrCIR† | ✗ | 33.76 | 51.86 | 28.11 | 49.43 | 35.70 | 57.32 | 32.52 | 52.87 |
| | SEIZE | ✗ | 33.04 | 53.22 | 30.93 | 50.76 | 35.57 | 58.64 | 33.18 | 54.21 |
| | **STT(Ours)** | ✗ | 29.95 | 48.66 | 20.21 | 42.95 | 31.70 | 53.21 | 27.28 | 48.27 |
| ViT-G/14 | Pic2Word | ✓ | 33.17 | 50.39 | 25.43 | 47.65 | 35.24 | 57.62 | 31.28 | 51.89 |
| | SEARLE | ✓ | 36.46 | 55.35 | 28.16 | 50.32 | 39.83 | 61.45 | 34.81 | 55.71 |
| | LinCIR | ✓ | 46.76 | 65.11 | 38.08 | 60.88 | 50.48 | 71.09 | 45.11 | 65.69 |
| | CIReVL | ✗ | 29.85 | 51.07 | 27.07 | 49.53 | 35.80 | 56.14 | 32.19 | 52.36 |
| | CIReVL* | ✗ | 31.65 | 49.07 | 23.90 | 43.13 | 32.53 | 53.19 | 29.36 | 48.46 |
| | CIReVL† | ✗ | 32.63 | 50.05 | 25.09 | 45.12 | 34.42 | 55.12 | 30.71 | 50.10 |
| | OSrCIR | ✗ | 38.65 | 54.71 | 33.02 | 54.78 | 41.04 | 61.83 | 37.57 | 57.11 |
| | OSrCIR* | ✗ | 36.56 | 55.45 | 30.69 | 53.25 | 40.13 | 61.30 | 35.79 | 56.67 |
| | OSrCIR† | ✗ | 37.39 | 56.92 | 30.59 | 53.50 | 39.72 | 61.04 | 35.79 | 57.15 |
| | SEIZE | ✗ | 43.60 | 65.42 | 39.61 | 61.02 | 45.94 | 71.12 | 43.05 | 65.85 |
| | **STT(Ours)** | ✗ | 39.48 | 56.59 | 35.04 | 56.74 | 42.86 | 64.95 | 39.12 | 59.43 |

Table 8: Performance comparison on GeneCIS datasets. **Both ViT-B and ViT-L are loaded from openai official weights, while ViT-G is loaded from openclip**.

| GeneCIS → | | | Focus Attribute | | | Change Attribute | | | Focus Object | | | Change Object | | | Average |
|---|---|---|---|---|---|---|---|---|---|---|---|---|---|---|---|
| Arch | Method | Train | R@1 | R@2 | R@3 | R@1 | R@2 | R@3 | R@1 | R@2 | R@3 | R@1 | R@2 | R@3 | R@1 |
| ViT-B/32 | SEARLE | ✓ | 18.9 | 30.6 | 41.2 | 13.0 | 23.8 | 33.7 | 12.2 | 23.0 | 33.3 | 13.6 | 23.8 | 33.3 | 14.4 |
| | CIReVL | ✗ | 17.9 | 29.4 | 40.4 | 14.8 | 25.8 | 35.8 | 14.6 | 24.3 | 33.3 | 16.1 | 27.8 | 37.6 | 15.9 |
| | OSrCIR | ✗ | 19.4 | 32.7 | 42.8 | 16.4 | 27.7 | 38.1 | 15.7 | 25.7 | 35.8 | 18.2 | 30.1 | 39.4 | 17.4 |
| | **STT(Ours)** | ✗ | 21.1 | 35.0 | 45.5 | 17.9 | 29.9 | 40.4 | 16.4 | 28.5 | 38.9 | 18.3 | 30.1 | 39.5 | 18.4 |
| ViT-L/14 | SEARLE | ✓ | 17.1 | 29.6 | 40.7 | 16.3 | 25.2 | 34.2 | 12.0 | 22.2 | 30.9 | 12.0 | 24.1 | 33.9 | 14.4 |
| | LinCIR | ✓ | 16.9 | 30.0 | 41.5 | 16.2 | 28.0 | 36.8 | 8.3 | 17.4 | 26.2 | 7.4 | 15.7 | 25.0 | 12.2 |
| | Context-I2W | ✓ | 17.2 | 30.5 | 41.7 | 16.4 | 28.3 | 37.1 | 8.7 | 17.9 | 26.9 | 7.7 | 16.0 | 25.4 | 12.7 |
| | CIReVL | ✗ | 19.5 | 31.8 | 42.0 | 14.4 | 26.0 | 35.2 | 12.3 | 21.8 | 30.5 | 17.2 | 28.9 | 37.6 | 15.9 |
| | OSrCIR | ✗ | **20.9** | 33.1 | 44.5 | 17.2 | 28.5 | 37.9 | 15.0 | 23.6 | 34.2 | 18.4 | 30.6 | 38.3 | 17.9 |
| | SEIZE | ✗ | 20.5 | 33.4 | 45.0 | 17.6 | 28.9 | 38.5 | 15.4 | 25.6 | 36.2 | 18.7 | 30.9 | 39.8 | 18.1 |
| | **STT(Ours)** | ✗ | 20.3 | 34.6 | 46.4 | 18.3 | 29.8 | 41.6 | 16.8 | 28.5 | 38.4 | 18.8 | 31.0 | 40.3 | 18.6 |
| ViT-G/14 | LinCIR | ✓ | 19.1 | 33.0 | 42.3 | 17.6 | 30.2 | 38.1 | 10.1 | 19.1 | 28.1 | 7.9 | 16.3 | 25.7 | 13.7 |
| | CIReVL | ✗ | 20.5 | 34.0 | 44.5 | 16.1 | 28.6 | 39.4 | 14.7 | 25.2 | 33.0 | 18.1 | 31.2 | 41.0 | 17.4 |
| | OSrCIR | ✗ | **22.7** | 36.4 | 47.0 | 17.9 | 30.8 | 42.0 | 16.9 | 28.4 | 36.7 | **21.0** | **33.4** | **44.2** | 19.6 |
| | SEIZE | ✗ | **22.9** | 36.2 | 47.3 | 18.6 | 31.4 | 42.7 | 18.2 | 28.8 | 37.6 | 19.6 | 33.0 | 43.5 | 19.8 |
| | **STT(Ours)** | ✗ | 21.9 | 36.4 | 47.9 | 19.6 | 31.9 | 42.8 | 20.2 | 30.3 | 39.6 | 19.7 | 33.2 | 43.4 | **20.4** |

## A.3 IMPACTS OF DIFFERENT MLLMS

Like previous works that employ MLLMs to analyze multimodal inputs and generate target descriptions, we specify Qwen2-VL-7B as the MLLM in earlier experiments. Here, we further explore the

Table 9: Performance comparison on CIRCO and CIRR datasets with various MLLMs.

| CIRCO + CIRR → | CIRCO | | | | CIRR | | | | | |
|---|---|---|---|---|---|---|---|---|---|---|
| Metric | mAP@k | | | | Recall@k | | | Recall$_{Subset}$@k | | |
| Method | k=5 | k=10 | k=25 | k=50 | k=1 | k=5 | k=10 | k=1 | k=2 | k=3 |
| Qwen-2B | 22.49 | 23.64 | 25.90 | 26.95 | 26.05 | 53.28 | 65.59 | 64.53 | 82.46 | 91.25 |
| Qwen-7B | 25.55 | 26.27 | 28.81 | 29.99 | 28.87 | 57.97 | 69.90 | 65.22 | 84.10 | 92.37 |
| LLaVA-Next (Mistral-7B) | 24.17 | 24.73 | 27.03 | 28.11 | 26.97 | 55.10 | 66.92 | 65.01 | 82.75 | 91.40 |
| GPT-4o(mini) | 25.68 | 26.50 | 29.16 | 30.30 | 28.59 | 58.13 | 69.99 | 66.15 | 84.98 | 92.86 |

Table 10: Ablation results on the transition and transportation modules. All results are conducted on CIRCO datasets with GPT-4o(mini).

| CIRCO + CIRR → | | CIRCO | | | |
|---|---|---|---|---|---|
| Strategy | | mAP@k | | | |
| Transition | Transportation | k=5 | k=10 | k=25 | k=50 |
| ✗ | ✗ | 35.49 | 37.05 | 40.02 | 41.28 |
| ✓ | ✗ | 37.50 | 39.10 | 42.24 | 43.50 |
| ✗ | ✓ | 36.61 | 38.10 | 41.11 | 42.38 |
| ✓ | ✓ | **38.93** | **40.14** | **43.18** | **44.46** |

performance of STT with different MLLMs. Specifically, we report the results on Qwen2-VL-2B, Qwen2-VL-7B, LLaVA-Next-7B, and GPT-4o(mini) in Tab. 9. The results show that our STT can be applied to MLLMs with different architectures and that the performance improves as the number of MLLM's parameters increases. This demonstrates the potential of STT in flexibility and scalability, as it serves as a plug-and-play pipeline that can seamlessly integrate with various MLLMs. Indeed, we observe that different MLLMs can lead to variations in the generated captions and thus impact retrieval results. This observation further supports our core motivation: rather than re-training or fine-tuning the large models, we aim to design a framework that maximizes retrieval effectiveness given any off-the-shelf MLLM.

In addition to Tab.4 that ablates each module on Qwen-7B, we also report the results with another MLLM GPT-4o(mini) in Tab.10. The ablations on two MLLMs can show the real efficiency of STT's modules: (1) *Strategic Synergy Over Raw MLLM Power*: The highest mAP@k values (e.g., 38.93 @k=5, 44.46 @k=50) occur when both Transition and Transportation are enabled. This indicates that STT's strength lies in its systematic orchestration of strategies rather than relying solely on MLLM capabilities. Even with the same MLLM (e.g., GPT-4o(mini)), disabling either strategy reduces performance (e.g., Transportation only yields 36.61 @k=5; Transition only yields 37.50 @k=5), confirming that STT actively improves task-specific reasoning. (2) *Modular Adaptability*: The results implies STT's strategies are architecture-agnostic. While the choice of MLLM impacts absolute performance, the framework's relative gains from Transition+Transportation synergy remain consistent.

## A.4  ADDITIONAL ABLATION EXPERIMENTS

### A.4.1  IMPACTS OF THE BIDIRECTIONAL DISTANCE.

To conduct a more comprehensive analysis of the impacts of The bidirectional distance, we supplemented experiments with STT under different backbones using CT distance and OT distance as alignment strategy in Tab. 11 and Tab. 12. The results show that CT outperforms OT, highlighting the advantages of bidirectional fine-grained alignment.

### A.4.2  IMPACTS OF CAPTION NUMBER AND AUGMENTATION VIEWS.

Moreover, for clarity, we have provided the specific values corresponding to Fig. 3 in the main text and supplemented the results of ablation experiments under different architectures, which can be found in Tab. 13. It is evident that compared to a single caption ($k$=1), multiple captions can provide richer multi-modal knowledge to better understand the implicit input, leading to more accurate descriptions.

Table 11: Ablation on CIRCO and CIRR datasets.

| CIRCO + CIRR → | | CIRCO | | | | CIRR | | | | | |
|---|---|---|---|---|---|---|---|---|---|---|---|
| Metric | | mAP@k | | | | Recall@k | | | Recall_Subset@k | | |
| Arch | Method | k=5 | k=10 | k=25 | k=50 | k=1 | k=5 | k=10 | k=1 | k=2 | k=3 |
| ViT-B/32 (OpenAI) | STT | 20.26 | 21.01 | 23.01 | 24.04 | 25.83 | 55.18 | 68.22 | 65.64 | 83.60 | 92.80 |
| | STT w/OT | 19.81 | 20.48 | 22.33 | 23.31 | 24.72 | 53.66 | 66.77 | 65.23 | 83.64 | 92.46 |
| ViT-B-32 (OpenCLIP) | STT | 28.21 | 28.99 | 31.31 | 32.53 | 32.56 | 62.10 | 73.86 | 70.00 | 86.60 | 94.87 |
| | STT w/OT | 23.94 | 24.79 | 27.01 | 28.09 | 31.37 | 60.97 | 72.92 | 69.54 | 85.76 | 94.27 |
| ViT-L/14 (OpenAI) | STT | 25.55 | 26.27 | 28.81 | 29.99 | 28.87 | 57.97 | 69.90 | 65.22 | 84.10 | 92.37 |
| | STT w/OT | 24.76 | 25.85 | 28.52 | 29.67 | 28.05 | 57.01 | 69.64 | 65.71 | 83.74 | 92.17 |
| ViT-L-14 (OpenCLIP) | STT | 32.31 | 33.33 | 36.32 | 37.49 | 35.04 | 65.57 | 76.41 | 71.52 | 88.00 | 94.65 |
| | STT w/OT | 30.23 | 31.22 | 34.13 | 35.2 | 34.39 | 64.48 | 76.17 | 71.98 | 88.34 | 94.68 |

Table 12: Performance on GeneCIS datasets.

| GeneCIS → | | Focus Attribute | | | Change Attribute | | | Focus Object | | | Change Object | | | Average |
|---|---|---|---|---|---|---|---|---|---|---|---|---|---|---|
| Arch | Method | R@1 | R@2 | R@3 | R@1 | R@2 | R@3 | R@1 | R@2 | R@3 | R@1 | R@2 | R@3 | R@1 |
| ViT-B/32 (OpenAI) | STT | 21.1 | 35.0 | 45.5 | 17.9 | 29.9 | 40.4 | 16.4 | 28.5 | 38.9 | 18.3 | 30.1 | 39.5 | 18.4 |
| | STT w/OT | 20.5 | 34.1 | 44.9 | 17.5 | 30.2 | 40.1 | 16.3 | 28.0 | 38.3 | 18.3 | 29.9 | 39.4 | 18.2 |
| ViT-B-32 (OpenCLIP) | STT | 20.8 | 33.6 | 44.2 | 17.6 | 29.4 | 39.7 | 17.4 | 30.7 | 40.4 | 19.6 | 33.6 | 44.2 | 18.9 |
| | STT w/OT | 20.2 | 33.4 | 43.8 | 17.1 | 29.4 | 39.0 | 17.0 | 29.7 | 39.9 | 20.0 | 33.4 | 43.8 | 18.6 |
| ViT-L/14 (OpenAI) | STT | 20.3 | 34.6 | 46.4 | 18.3 | 29.8 | 41.6 | 16.8 | 28.5 | 38.4 | 18.8 | 31.0 | 40.3 | 18.6 |
| | STT w/OT | 20.6 | 34.5 | 45.8 | 18.0 | 29.2 | 40.3 | 16.8 | 27.9 | 38.2 | 18.8 | 30.2 | 40.1 | 18.5 |
| ViT-L-14 (OpenCLIP) | STT | 20.5 | 33.8 | 44.2 | 18.1 | 29.0 | 40.2 | 18.5 | 29.4 | 39.1 | 19.9 | 32.9 | 42.8 | 19.3 |
| | STT w/OT | 20.3 | 33.4 | 44.4 | 17.6 | 28.8 | 39.9 | 17.9 | 28.6 | 38.1 | 19.6 | 33.1 | 42.3 | 18.9 |

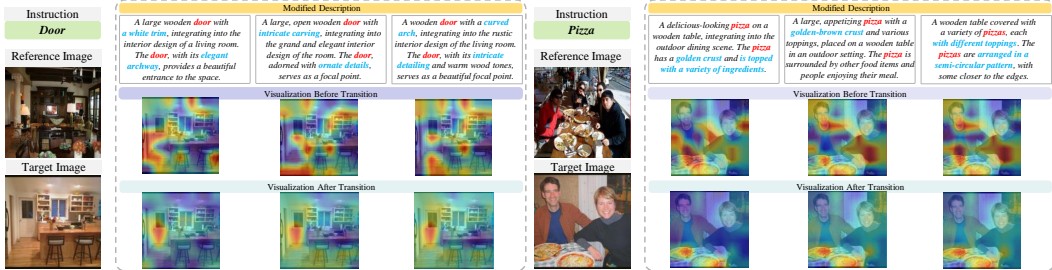

Figure 6: Visualization of the GeneCIS dataset on the 'Focus Object' task. Heatmaps before and after the transition on target image are shown. Captions generated by MLLMs often contain irrelevant visual noise (blue text), while the STT model effectively suppresses such noise and highlights the correct focus object (red text).

### A.4.3 HYPER-PARAMETERS STUDY

We report a sensitivity analysis of $\alpha$ in Tab.14. The results show that STT exhibits moderate sensitivity to $\alpha$, with performance being non-monotonic. Specifically, values in the range of $0.3 - 0.5$ yield optimal results, while overly small or large values degrade performance. This confirms the effectiveness of treating modification as a transition vector, as it helps mitigate biases between MLLM-generated captions and images. For practical use, in accuracy-critical tasks (e.g., CIRCO), we suggest $\alpha \leq 0.5$ to avoid over-modification; In recall-critical tasks (e.g., CIRR), starting with $\alpha = 0.4$ is reasonable. For new datasets, a grid search within $[0.3, 0.5]$ could be conducted, selecting the optimal $\alpha$ based on validation performance tailored to the application's specific needs.

### A.5 MORE VISUALIZATION

For a more comprehensive qualitative analysis, we present the visualization results of GeneCIS datasets about the task of focus in Fig. 6. It illustrated that the original generated descriptions indeed introduce visual noise while our STT often focus on the correct object, leading to higher CIR performance.

Table 13: Ablation study on CIRCO and CIRR datasets with different number of image augmentation on CLIP-B/32 and fix the number of description to 5.

| CIRCO + CIRR → | CIRCO | | | | CIRR | | | | | |
|---|---|---|---|---|---|---|---|---|---|---|
| Metrics | mAP@k | | | | Recall@k | | | Recall$_{Subset}$@k | | |
| Num | k=5 | k=10 | k=25 | k=50 | k=1 | k=5 | k=10 | k=1 | k=2 | k=3 |
| 1 | 19.73 | 19.89 | 21.68 | 22.63 | 25.16 | 53.59 | 66.46 | 63.88 | 82.87 | 92.19 |
| 5 | 20.19 | 20.84 | 22.70 | 23.73 | 25.25 | 54.00 | 67.40 | 64.46 | 83.61 | 92.39 |
| 10 | 20.26 | 21.01 | 23.01 | 24.04 | 25.83 | 55.25 | 68.22 | 65.64 | 83.60 | 92.80 |
| 25 | 20.60 | 21.37 | 23.55 | 24.55 | 25.64 | 55.45 | 68.87 | 65.71 | 84.41 | 92.46 |
| 50 | 21.01 | 21.62 | 23.74 | 24.79 | **26.15** | **55.78** | **69.16** | **66.17** | **84.74** | 93.06 |
| 100 | **21.96** | **22.51** | **24.60** | **25.62** | 25.67 | 55.69 | 69.08 | 65.45 | 84.36 | **93.08** |

Table 14: Sensitivity analysis of $\alpha$ on Qwen2-VL-7B and ViT-B/32 on CIRCO and CIRR datasets (default $\alpha = 0.45$ in our main manuscript).

| CIRCO + CIRR → | CIRCO | | | | CIRR | | | | |
|---|---|---|---|---|---|---|---|---|---|
| Metrics | mAP@k | | | | Recall@k | | | Recall$_{Subset}$@k | |
| $\alpha$ value | k=5 | k=10 | k=25 | k=50 | k=1 | k=5 | k=10 | k=1 | k=2 |
| 0.1 | 18.37 | 19.09 | 20.77 | 21.76 | 23.28 | 49.98 | 62.36 | 64.05 | 83.21 |
| 0.2 | 19.74 | 20.49 | 22.34 | 23.32 | 24.63 | 52.46 | 65.45 | 64.89 | 83.40 |
| 0.3 | 21.71 | 22.36 | 24.33 | 25.26 | 25.35 | 54.12 | 67.28 | 65.49 | 83.74 |
| 0.4 | 20.73 | 21.37 | 23.33 | 24.37 | 25.81 | 55.37 | 68.34 | 65.23 | 83.67 |
| 0.45 | 20.26 | 21.01 | 23.01 | 24.04 | 25.83 | 55.25 | 68.22 | 65.64 | 83.60 |
| 0.5 | 21.47 | 22.47 | 24.46 | 25.46 | 26.02 | 55.45 | 68.58 | 64.82 | 83.49 |
| 0.6 | 19.77 | 20.45 | 22.55 | 23.48 | 25.62 | 55.40 | 68.22 | 63.64 | 83.13 |
| 0.7 | 19.05 | 20.18 | 22.11 | 23.20 | 25.11 | 54.65 | 68.22 | 63.62 | 82.68 |

## A.6 STT In-Context Learning Details

We utilize an in-context learning method in Fig. 7. To achieve ZS-CIR, each sample uses the same placeholder "`<image_url>`" instead of an actual reference image URL. By providing several example outputs, the model is able to understand the required reasoning process without an actual reference image. This approach ensures efficient reasoning in a zero-sample setting. Each text requires the model to focus on a specific object and provide a detailed description. This helps the model understand the key elements in the image and how they relate to each other. We use uniform placeholders `<image_url>` and `<reference_image_url>` to ensure that the input and output formats are consistent for easy model processing.

## A.7 Further Comparison with SEIZE

We observe that both SEIZE Yang et al. (2024a) and our STT generate multiple captions and apply the semantic transition process. However, these two models are different from each other in terms of caption generation, semantic editing strategy, and retrieval score calculation: (1) **Two-Stage Generation vs. One-Stage Generation**: SEIZE first generates $N$ captions for the reference image using a captioner and then modifies them according to the input modification text via an LLM. In contrast, our STT directly employs an MLLM to generate $N$ captions for the composed input, eliminating information loss from two-stage approaches. Moreover, the efficiency comparison in Tab. 5 shows that two-stage generation methods are time-consuming, which may limit their applicability in real-time scenarios. (2) **Point-to-Point vs. Set-to-Set**: SEIZE represents the final global caption feature by employing the average pooling on captions, and then measures similarity with candidates via cosine similarity. Our STT, however, models the captions as a discrete distribution and then develops a transportation-aware set-to-set metric to calculate the distances. (3) **Similarity Space vs. Embedding Space**: SEIZE refines the final retrieval score by directly changing the cosine score. Our STT aims to refine the generated captions in the CLIP embedding space.

---

**Example 1**

*<Input>*

{

"**Reference Image 1**": <image_url>, *(each sample uses the same placeholder "<image_url>" instead of an actual image URL)*

"**Text modification 1**": "Instruction: Focus cardboard and its arrangement with other objects in the image. Describe cardboard with details. **Edited Description**: A flattened piece of cardboard placed at the base of a disorganized pile. The cardboard, with its worn and creased surface, shows frayed edges and slight tears, indicating frequent use or exposure to the elements. It serves as a foundation for the objects stacked above, including bright blue garbage bags, a broken white plastic chair, and parts of a large appliance."

"**Reference Image 2**": <image_url>, *(each sample uses the same placeholder "<image_url>" instead of an actual image URL)*

"**Text modification 2**": "Instruction: Focus wood floor and its arrangement with other objects in the image. Describe wood floor with details. **Edited Description**: A polished wood floor with a warm, rich tone that adds to the cozy ambiance of the room. The floor's natural grain is partially covered by a soft, beige area rug placed in the center, creating a harmonious balance between texture and color. Surrounding the rug, the floor extends under the dark brown sofa and armchair, complementing their earthy tones. The floor seamlessly integrates with the room's arrangement, supporting the furniture and decor, such as the side tables and potted plants, enhancing the overall inviting atmosphere."

"**Reference Image 3**": <reference_image_url_1>,

"**Text modification 3**": " Instruction: Focus **<relative_caption>** and its arrangement with other objects in the image. Describe **<relative_caption>** with details. **Edited Description**: "

}

*<Response>*

{

"Target Image Description": "A large wooden door with a white trim, integrating into the interior design of a living room. The door, with its elegant archway, provides a beautiful entrance to the space."

}

---

**Example 2**

*<Input>*

{

"**Reference Image 1**": <image_url>, *(each sample uses the same placeholder "<image_url>" instead of an actual image URL)*

"**Text modification 1**": "Instruction: Focus cardboard and its arrangement with other objects in the image. Describe cardboard with details. **Edited Description**: A flattened piece of cardboard placed at the base of a disorganized pile. The cardboard, with its worn and creased surface, shows frayed edges and slight tears, indicating frequent use or exposure to the elements. It serves as a foundation for the objects stacked above, including bright blue garbage bags, a broken white plastic chair, and parts of a large appliance."

"**Reference Image 2**": <image_url>, *(each sample uses the same placeholder "<image_url>" instead of an actual image URL)*

"**Text modification 2**": "Instruction: Focus wood floor and its arrangement with other objects in the image. Describe wood floor with details. **Edited Description**: A polished wood floor with a warm, rich tone that adds to the cozy ambiance of the room. The floor's natural grain is partially covered by a soft, beige area rug placed in the center, creating a harmonious balance between texture and color. Surrounding the rug, the floor extends under the dark brown sofa and armchair, complementing their earthy tones. The floor seamlessly integrates with the room's arrangement, supporting the furniture and decor, such as the side tables and potted plants, enhancing the overall inviting atmosphere."

"**Reference Image 4**": <reference_image_url>,

"**Text modification 4**": " Instruction: Focus **<relative_caption>** and its arrangement with other objects in the image. Describe **<relative_caption>** with details. **Edited Description**: "

}

*<Response>*

{

"Target Image Description": "A delicious-looking pizza on a wooden table, integrating into the outdoor dining scene. The pizza has a golden crust and is topped with a variety of ingredients."

}

---

Figure 7: Examples of our in-context learning on GeneCIS dataset. Each sample uses the same placeholder "`<image_url>`" instead of an actual reference image URL.

.

## A.8 LIMITATIONS AND FUTURE WORK

Although our method achieves strong performance, there remain several directions for future exploration. First, when the query image depicts a complex scene involving multiple objects or relationships, and the accompanying modification text provides insufficient detail, our STT may focus on the wrong or ambiguous object, leading to unexpected captions. This limitation is consistent with issues observed in prior CIReVL Karthik et al. (2023) and OSrCIR Tang et al. (2024a) models. Moreover, current benchmarks suffer from a false-negative problem. As noted in Liu et al. (2021), each (reference image, modification) pair in FashionIQ can correspond to multiple valid target images, yet only one is annotated as ground truth. Consequently, semantically correct retrieval results may be unfairly penalized under existing evaluation protocols. We leave these challenges as promising directions for future research.

## A.9 THE USE OF LARGE LANGUAGE MODELS

In this work, Large Language Models (LLMs) were used exclusively for language polishing and spelling correction.

