# OpenReview forum: "STT: Towards Training-Free Zero-Shot Composed Image Retrieval via Semantic Transition and Transportation"
_ICLR.cc/2026/Conference — ICLR 2026 Conference Withdrawn Submission_

### Official Review · Reviewer_At5H · 2025-10-31

**Soundness:** 3
**Presentation:** 2
**Contribution:** 2
**Rating:** 2
**Confidence:** 5

**Summary:**

The paper proposes a training free STT framework for zero shot composed image retrieval that targets the noise introduced by MLLM generated captions and the coarse point to point matching used in prior work. The authors first view multiple generated descriptions as a discrete text distribution and perform a semantic transition guided by the modification text, thereby alleviating the impact of noise. On the visual side, the target image is expanded into a multi-view distribution through augmentations. A bidirectional transport alignment is then applied to enforce mutual coverage between textual variants and visual views, leading to finer-grained retrieval.

**Strengths:**

1. The paper presents an MLLM guided semantic transition that helps reduce noise from generated captions.
2. The work adopts a bidirectional transport alignment over discrete text and image views to enable finer grained cross modal matching.
3. Experiments on several CIR benchmarks show consistent improvements over existing training free baselines.

**Weaknesses:**

1. The semantic transition module essentially adds the modification text as a semantic increment to the generated captions, which makes its core idea close to the semantic increment strategy in SEIZE, so its core contribution  insufficiently novel.
2. One of the core contributions is the bidirectional transport module, but the paper does not clearly demonstrate its advantage; in Table 4 it is compared only against an unclear baseline, and the appendix does not provide different point to point settings, which makes it hard to see the actual gain brought by this component.
3. The paper uses inconsistent terminology for the main components, sometimes referring to them as transition and transportation, sometimes as semantic transition and transportation, and elsewhere as bidirectional transport distance.

**Questions:**

1. It is unclear whether the transport module is ablated by directly removing it, as the paper does not clearly specify the corresponding comparison baseline, nor whether it outperforms direct point to point matching.
2. The paper does not specify the concrete image augmentation strategies used for the target samples, and it is unclear whether different augmentation choices would lead to different retrieval performance, for example when a particular crop happens to capture a crucial visual detail.
3. As shown in Figure 5, incorporating instruction signals helps the model focus on key objects on GeneCIS, but this effect is demonstrated only on that dataset, and it is unclear whether the same benefit holds for CIRR, CIRCO or other benchmarks with richer modification texts.
4. Is there a plan to open source the code? This is very important for reproducing the results of the paper. If the reviewers can address the weaknesses and questions raised, I am willing to raise the score.

---

### Official Review · Reviewer_zTfG · 2025-11-01

**Soundness:** 2
**Presentation:** 3
**Contribution:** 2
**Rating:** 4
**Confidence:** 4

**Summary:**

This paper introduces STT, a resampling method that leverages Conditional Transport to improve training-free ZS-CIR. The authors aim to address noisy target captions that inherit irrelevant details from the reference image and brittle point-to-point alignment between a single caption and an image. Experiments on CIRR, CIRCO, Fashion-IQ, and GeneCIS show competitive results across many settings and lower test-time latency than existing LLM-based methods.

**Strengths:**

1.	The writing of this paper is clear.
2.	The idea of this paper is easy to understand.
3.	It is reasonable to leverage Conditional Transport method for ZS-CIR.

**Weaknesses:**

1.	Limited novelty. Resampling for more comprehensive image understanding in ZS-CIR has been explored [1,2]. The proposed method appears incremental: It resamples target-image information via augmentation (e.g., random cropping) and introduces a Conditional Transport approach to achieve many-to-many alignment. However, I would argue that many-to-many alignment does not address the core challenge of CIR (e.g., understanding user intent), which may explain the limited performance gains. Consequently, I am concerned that the novelty does not provide sufficient new insight for the ICLR community.
2.	Limited technology contribution. The target-image augmentation (random crop) follows PrediCIR [3], which the authors overlook. Moreover, the one-stage, training-free, in-context setup builds on OSrCIR [4], and diverse caption generation echoes LDRE [1]. As a result, the main technical contribution appears to be the Conditional Transport formulation, i.e., Eq.(6), for many-to-many alignment in ZS-CIR.
3.	Questionable motivation. In my knowledge, the predicted caption is a concise textual query for CLIP retrieval (often ≤40 tokens), so it may not inherit many irrelevant details. Furthermore, many-to-many alignment may still miss the central challenge, which aims for user intention understanding. It might cause the limited performance improvement of this paper.
4.	Concerns of the generalizability. The authors claim their method aims to improve the “generation-then-retrieval” pipeline in ZS-CIR without sufficient experiment to support the claim. They only evaluate their method in their pipeline. It makes me concern about the generalizability of STT.
5.	Missing implementary details. The pipeline include many part and the code are not given. For example, there are not sufficient details of how the augmentation step perform in the Transition module.
6.	Concerns about the hallucination issues. The output of STT is a MLLM generated results without CoT process, which make me concern the hallucination problem. Such issues could significantly impact the retrieval results. A more detailed analysis of hallucination risks is needed.
7.	Need more analysis experiments. For example, have a visualization experiments to show the benefit of the Alignment module which leverage CT framework for many to many alignment.
8.	Insufficient ablation studies. For example, What is the performance of the pipeline with CoT process ? What is the performance with other better MLLM (i.e., GPT-5, GPT-4o, Qwen-VL-72B)? What is the influence of other cropping method rather than random crop?

Overall, the novelty and technology contribution appear limited, and the motivation needs to improve. Key implementation details and hallucination analysis are missing. Therefore, I have given the Weak Reject recommendation.

References

[1] Yang Z, Xue D, Qian S, et al. Ldre: Llm-based divergent reasoning and ensemble for zero-shot composed image retrieval[C]//Proceedings of the 47th International ACM SIGIR conference on research and development in information retrieval. 2024: 80-90.

[2] Yang Z, Qian S, Xue D, et al. Semantic editing increment benefits zero-shot composed image retrieval[C]//Proceedings of the 32nd ACM International Conference on Multimedia. 2024: 1245-1254.

[3] Tang Y, Yu J, Gai K, et al. Missing target-relevant information prediction with world model for accurate zero-shot composed image retrieval[C]//Proceedings of the Computer Vision and Pattern Recognition Conference. 2025: 24785-24795.

[4] Tang Y, Zhang J, Qin X, et al. Reason-before-retrieve: One-stage reflective chain-of-thoughts for training-free zero-shot composed image retrieval[C]//Proceedings of the Computer Vision and Pattern Recognition Conference. 2025: 14400-14410.

**Questions:**

1.	What is the performance of STT in other training-free CIR pipelines (e.g., CIReVL, LDRE, OSrCIR)?
2.	Have you analyzed the benefits of Conditional Transport in the Alignment module for CIR?
3.	Can you provide the performance with more training-free method with the proposed STT?
4.	What is the performance of the pipeline with CoT process ?
5.	What is the performance with other better MLLM (i.e., GPT-5, GPT-4o, Qwen-VL-72B)?
6.	What is the influence of other cropping method rather than random crop?

---

### Official Review · Reviewer_gBvc · 2025-11-01

**Soundness:** 3
**Presentation:** 3
**Contribution:** 3
**Rating:** 4
**Confidence:** 3

**Summary:**

The paper identifies two key limitations of existing training-free, generation-based zero-shot composed image retrieval (ZS-CIR) methods: (1) erroneous emphasis on target caption generation biased toward the reference image, and (2) a restrictive point-to-point retrieval formulation. To address these issues, the authors propose STT (Semantic Transition and Transportation), which introduces a modification-transition strategy and reformulates the retrieval task as a set-to-set alignment problem, modeled as a bidirectional transport problem. Experimental results demonstrate the strong effectiveness of each proposed component across diverse scenarios and benchmarks. However, further clarification is needed regarding the method’s efficiency and the process for obtaining image augmentations.

**Strengths:**

1. The identified problems  (the erroneous emphasis on target caption generation and the point-to-point retrieval formulation) are well-motivated and clearly justified.
2. The paper is well-written, well-structured, and easy to follow for novice readers.
3. The proposed method appears well-motivated and effective, with experiments and analyses that convincingly support the reported improvements.
4. Compared to the conventional two-stage generation procedure, the proposed one-stage formulation offers efficiency advantages.

**Weaknesses:**

1. Efficiency analysis and image augmentation. In the efficiency analysis, it is unclear whether the computation time for generating target image augmentations is included. More clarification is needed regarding how the augmented images are obtained in formulation (5). The feature extraction phase for target image augmentations is likely to be computationally intensive, and if such augmentations are required for the entire database, the overall cost could be significantly higher than that of previous methods. This may become infeasible for large-scale datasets. Therefore, it is important to clarify whether this cost is included in the reported efficiency and how the proposed “efficient” implementation handles this aspect. This point could be critical for evaluating the practical feasibility of the method.

2. As shown in Table 4, the baseline (row 1) already appears to outperform CIReVL, which raises concerns about the fairness of the comparison with prior state-of-the-art methods. For a fair evaluation, it is important to ensure that all methods are assessed under the same experimental conditions (particularly using the same underlying MLLMs) to avoid overestimating the performance gains attributed to the proposed method.

3. Alternative set-to-set alignment formulations. It would be helpful to explore or at least discuss possible alternatives for the set-to-set alignment objective (Ablation studies comparing different loss formulations beyond L_bi)
​
4. Additional analysis or discussion on the sensitivity of the hyperparameter alpha in the transition phase

**Questions:**

Wrote above

---

### Official Review · Reviewer_PZS9 · 2025-11-04

**Soundness:** 3
**Presentation:** 3
**Contribution:** 2
**Rating:** 2
**Confidence:** 4

**Summary:**

This paper proposes a semantic transition and transportation (STT) framework for the zero-shot composed image retrieval task, which refines the inferred composed caption through a transition vector in the embedding space and makes it closer to the target image.

**Strengths:**

1. The papre is well written and easy to follow.
2. The illustrations are well oragnized and clearly demonstrate the motivation and architecture of the proposed method.

**Weaknesses:**

1. The effectiveness of the proposed method needs further validation, since the performance of STT on the Fashion-IQ dataset is severely inferior to the comparison methods.
2. The core querying-transition-alignment precodure of the proposed STT method is not novel. Could the author provide theoretical insights of STT to strengthen the novelty?

**Questions:**

1. The performance on the Fashion-IQ dataset is inferior to the comparison methods, and the improvements are not consistent on the CIRR and GenCIS datasets as well.
2. More theoritical insights of the proposed method should be provided to strengthen the novelty.
3. More details of efficiency evaluation results such as memory usage and GFLOPs should be provided to improve the completeness of the paper.

---

### Note · Authors · 2025-11-13

I have read and agree with the venue's withdrawal policy on behalf of myself and my co-authors.